# Immunoinformatics Design and In Vivo Immunogenicity Evaluation of a Conserved CTL Multi-Epitope Vaccine Targeting HPV16 E5, E6, and E7 Proteins

**DOI:** 10.3390/vaccines12040392

**Published:** 2024-04-09

**Authors:** Ni Guo, Zhixin Niu, Zhiling Yan, Weipeng Liu, Lei Shi, Chuanyin Li, Yufeng Yao, Li Shi

**Affiliations:** 1Yunnan Key Laboratory of Vaccine Research & Development on Severe Infectious Disease, Institute of Medical Biology, Chinese Academy of Medical Sciences & Peking Union Medical College, Kunming 650118, China; guoni@student.pumc.edu.cn (N.G.); niuzhixinamber@163.com (Z.N.); liuweipeng@imbcams.com.cn (W.L.); chuanyinli@imbcams.com.cn (C.L.); 2Department of Gynaecologic Oncology, Peking University Cancer Hospital Yunnan & Yunnan Cancer Hospital & The Third Affiliated Hospital of Kunming Medical University, Kunming 650118, China; yanzhiling2021@126.com; 3Department of Immunogenetics, Institute of Medical Biology, Chinese Academy of Medical Sciences & Peking Union Medical College, Kunming 650118, China; sansan33@imbcams.com.cn

**Keywords:** human papilloma virus type 16 (HPV16), immunoinformatics, CTL epitope prediction, multi-epitope vaccine design, tumor immunotherapy

## Abstract

Human papillomavirus type 16 (HPV16) infection is responsible for more than 50% of global cervical cancer cases. The development of a vaccine based on cytotoxic T-lymphocyte (CTL) epitopes is a promising strategy for eliminating pre-existing HPV infections and treating patients with cervical cancer. In this study, an immunoinformatics approach was used to predict HLA-I-restricted CTL epitopes in HPV16 E5, E6, and E7 proteins, and a set of conserved CTL epitopes co-restricted by human/murine MHCs was screened and characterized, with the set containing three E5, four E6, and four E7 epitopes. Subsequently, the immunogenicity of the epitope combination was assessed in mice, and the anti-tumor effects of the multi-epitope peptide vaccine E5E6E7pep11 and the recombinant protein vaccine CTB-Epi11E567 were evaluated in the TC-1 mouse tumor model. The results demonstrated that mixed epitope peptides could induce antigen-specific IFN-γ secretion in mice. Prophylactic immunization with E5E6E7pep11 and CTB-Epi11E567 was found to provide 100% protection against tumor growth in mice. Moreover, both types of the multi-epitope vaccine significantly inhibited tumor growth and prolonged mouse survival. In conclusion, in this study, a multi-epitope vaccine targeting HPV16 E5, E6, and E7 proteins was successfully designed and evaluated, demonstrating potential immunogenicity and anti-tumor effects and providing a promising strategy for immunotherapy against HPV-associated tumors.

## 1. Introduction

Persistent infection with human papillomaviruses (HPVs) contributes to the global cervical cancer burden [1,2]. More than 200 HPV types have been identified and further classified into high-risk (HR) and low-risk (LR) HPV based on their association with cervical cancer. HPV16 represents the most significant high-risk human papillomavirus (HR-HPV), accounting for more than half of cervical cancer cases worldwide [2,3]. Although current vaccines have demonstrated significant efficacy in preventing cervical cancer, they are unable to clear pre-existing HPV infections. Developing therapeutic HPV vaccines presents numerous challenges, including the rapid identification of viable target antigens, issues regarding cross-protection due to antigenic diversity and antigenic variation, immune evasion within the tumor microenvironment, as well as the assessment of vaccine immunogenicity and long-term safety [4,5].

Cytotoxic T lymphocytes (CTLs) play an essential role in the host’s defense against viral infections. When a cell is infected by viruses, fragments of viral proteins are presented on the host cell surface via major histocompatibility complex (MHC) class I molecules. Cytotoxic T cells recognize these protein fragments—also known as epitopes—in conjunction with MHC class I molecules, activating specific CD8^+^ T cytotoxic T cell or CD4^+^ helper T cell responses, thereby facilitating virus and infected cell clearance [6,7]. Epitope-based vaccines leverage the functionality of CTLs in clearing viruses and offer several advantages compared to traditional vaccines. Firstly, they can elicit a broader CTL response by being engineered to carry multiple epitopes from various parts of the pathogen. Secondly, they can be more rapidly produced, thus enabling rapid adaptation in response to viral mutations and epidemics. Furthermore, this type of vaccine contains only critical immunogenic regions of the pathogen to stimulate effective immunity, excluding other components that might cause adverse effects [8].

In studies on HPV16-associated cervical cancer vaccines, researchers are endeavoring to develop multi-epitope vaccines that target viral oncoproteins co-expressed by tumor cells. The E6 and E7 proteins, the principal oncogenic proteins of HR-HPVs, are persistently expressed in tumors, making them constant immunological targets and prime candidates for vaccine development [9]. The results of some studies have indicated that although the E5 protein has reduced oncogenic potential, it can enhance cellular receptor activity and play a role in certain HPV-related cancers [10,11,12].

Vaccine developers have long been devoted to designing vaccines with precise, efficient, and extensive immune responses more quickly and accurately. Due to co-evolution with humans, HPV16 has evolved into four variant lineages (A, B, C, and D) and sixteen sublineages: A1–A3 (EUR), A4 (As), B1–B4 (AFR-1), C1–C4 (AFR-2), D1 (NA), D2 (AA2), D3 (AA1), and D4 (AA1) [13]. The results of several large-scale case–control studies have shown that HPV variant lineages are associated with the disease risk, persistence, and resulting histological types of lesions in different populations [14,15,16]. In our previous study, we found significant differences in the distribution of mutations in the LCR, E1, and E7 genes of HPV16, and these mutations showed different amounts and frequencies in different HPV16 lineages [17,18,19]. These findings suggest that the variation in and mutation of HPV16 may contribute to the immune evasion of the virus and the establishment of persistent infection, leading to cervical cancer. Therefore, selecting highly conserved CTL epitopes can aid the vaccine in providing protection against diverse HPV16 lineages.

In recent years, immunoinformatics tools have become increasingly important in vaccine design. Such tools enable researchers to efficiently predict potential CTL epitopes. Using immunoinformatics tools for the prediction and analysis of CTL epitopes in antigens can guide vaccine development toward greater precision [20]. Accordingly, in the current study, we initially obtained the conserved CTL epitopes of HPV16 E5, E6, and E7 proteins capable of binding to high-frequency distributed HLA-class I molecules using immunoinformatics predictions, constructed a combined mixed-peptide vaccine and a recombinant protein vaccine incorporating these epitopes, and assessed their immunogenicity and protective efficacy against tumors in mice.

## 2. Materials and Methods

### 2.1. Sequence Retrieval of the HPV16 Genome and E5, E6, and E7 Proteins

Sixty-eight distinct HPV16 whole-genome sequences classified into sixteen sublineage variants were obtained from the PaVE database (https://pave.niaid.nih.gov/explore/variants/variant_genomes, accessed on 10 September 2022) and the data included in a previous study [13,21]. With reference to the accession numbers of these isolates, the corresponding E5, E6, and E7 protein IDs were retrieved from the NCBI (https://ncbi.nlm.nih.gov/, accessed on 10 September 2022), and the protein sequences were downloaded in FASTA format. The lineage (traditionally by geographical region) and sublineage (by numeric–alphabetic nomenclature), strain name, genome ID number, and protein ID number are summarized in Appendix A.

### 2.2. Immunoinformatics Prediction

The T-Cell Epitope Prediction Tools from the Immune Epitope Database (IEDB, version 2.26) analysis resource (http://tools.iedb.org/mhci/, accessed on 17 September 2022) were used for protein epitope prediction analysis. NetMHCPan 4.1 EL was chosen as the default optimal prediction method, which can be used to comprehensively evaluate the ability of a peptide to bind MHC molecules and the likelihood that the peptide will be naturally processed and presented to derive a score for each epitope. The E5, E6, or E7 protein sequence of the HPV16 Ref isolate (NC_001526) was used for prediction. Based on the recommendations of the IEDB, only frequently occurring HLA class I alleles, which occur in at least 1% of the global population, were selected for analysis. The respective genotypic frequency in the global population was calculated using data from studies conducted around the globe provided by the Allele Frequency Net Database (AFND, http://allelefrequencies.net/, accessed on 20 September 2022). The peptide lengths were limited to 8–11 amino acids for the selected HLA class I. Mouse MHC-I-(H-2b-)-restricted CTL epitope prediction was performed using the same tools and parameters described above.

### 2.3. Conserved CTL Epitope Identification and Population Coverage Analysis

The conservancy analysis tool (http://tools.iedb.org/conservancy/, accessed on 25 September 2022) provided by the IEDB (version 2.26) was used to assess the conservancy of the epitopes. FASTA protein sequences of E5, E6, and E7 derived from 68 HPV16 reference isolates were entered after the removal of duplicate protein sequences to calculate the degree of conservancy of selected epitopes across all HPV16 sublineages. Population coverage for each candidate epitope was analyzed using the IEDB population coverage calculation tool (http://tools.iedb.org/population/, accessed on 26 September 2022), which is based on the allele frequencies of individuals from 115 countries and 21 different ethnic groups (divided into 16 different geographical regions) provided by the HLA Allele Frequency Database (2020 update). The calculation of coverage was carried out based on the world population, and we selected the MHC I separate calculation option for CTL responses.

### 2.4. Immunogenicity, Antigenicity and Toxicity Evaluation of Candidate Epitopes

The immunogenicity of each candidate’s conserved CTL epitope was predicted using the class I immunogenicity analysis tool provided by the IEDB (http://tools.iedb.org/immunogenicity, accessed on 28 September 2022). The properties of the amino acids and their position in the peptide segment were used as the main parameters to evaluate the immunogenicity of the peptide MHC (pMHC) complex. Protective antigenic or non-antigenic predictions were performed for each epitope peptide using the VaxiJen v2.0 server. The AllerTOP v.2 server (https://www.ddg-pharmfac.net/AllerTOP/, accessed on 28 September 2022) was used to predict and identify potential allergens in candidate epitope peptides. The ToxinPred2 server (https://webs.iiitd.edu.in/raghava/toxinpred2/batch.html, accessed on 28 September 2022) was used to predict the toxicity of each candidate peptide.

### 2.5. Preparation of the Mixed-Peptide Vaccine

The candidate epitope peptides were chemically synthesized by GeneScript (Nanjing, China) with ≥95% purity. Each peptide was dissolved to its highest attainable concentration and subsequently diluted to a uniform 1 mg/mL using phosphate-buffered saline (PBS, 0.01 M, pH = 7.4). These preparations were then mixed with the CpG ODN 1826 adjuvant to generate a composite peptide vaccine designated as E5E6E7pep11. The final formulation of E5E6E7pep11 consisted of a concentration of 5 μg of each peptide and 20 μg of the adjuvant per 100 μL volume.

### 2.6. Construction of Recombinant Protein Vaccines

The conserved CTL epitope was linearly linked through the use of an “AAY” spacer sequence, and a cholera toxin B subunit (CTB) was conjugated at the N-terminus to function as an intramolecular adjuvant, thereby enhancing the immunogenicity of the multi-epitope vaccine construct.

The gene sequence for recombinant protein CTB-Epi11E567 was codon optimized using GeneOptimizer™ (Thermo Fisher Scientific, Waltham, MA, USA) and then synthesized by Sangon Biotech (Shanghai, China). To construct the expression vector, the gene was amplified by polymerase chain reaction (PCR) with the following primer pairs: forward primer 5′-accctcgagggatccgaattcATGATCAAACTGAAATTCGGC-3′ and reverse primer 5′-caggtcgacaagcttgaattcCAGGGTGCCCATCAGCAG-3′. Afterward, the amplified fragments with the homology arm of the linearized vector were cloned into a linearized pCold-TF vector digested by EcoR I using 2× Hieff Clone^®^ Universal Enzyme Premix (Yeasen, Shanghai, China), and the expression vector pCold-TF-CTB-Epi11E567 was constructed.

The pCold-TF-CTB-Epi11E567 was transformed into competent *E. coli* (BL21). These cells were then cultured in LB medium containing 100 μg/mL of ampicillin at 37 °C with shaking until an OD600 of approximately 0.6 was reached, at which point IPTG was added to a final concentration of 0.5 mM to induce expression. The culture was then incubated at 15 °C to promote protein expression. After 24 h of expression, cells were harvested via centrifugation and resuspended in an ultrasonic buffer for lysis via sonication in an ice bath. The cell lysates were centrifuged at 12,000× *g* for 20 min at 4 °C to separate the supernatant containing the soluble target protein. The target protein, featuring a 6 × His-TF tag, was then purified via affinity chromatography using BeyoGold™ His-tag Purification Resin (Beyotime Biotechnology, Shanghai, China). Subsequently, the 6 × His-TF tag protein was removed via cutting with HRV 3C Protease (Takara Bio, Shiga, Japan) and passage through BeyoGold™ His-tag Purification Resin. The protein was further validated through 10% SDS-PAGE and Coomassie Blue staining, and its concentration was quantified using the BCA method (Thermo Fisher Scientific, Waltham, MA, USA). The recombinant protein was then filter sterilized through a 0.22 μm membrane and stored at −80 °C.

### 2.7. Physiochemical Property Evaluation

The ProtParam tool provided by Expasy 3.0 (http://web.expasy.org/protparam, accessed on 1 March 2023) was used to predict the physicochemical properties of the recombinant protein CTB-Epi11E567 and Epi11E567. Parameters including molecular weight (MW), theoretical isoelectric point, aliphatic index, instability index, and grand average of hydropathicity (GRAVY) were computed.

### 2.8. Cell Line and Mice

The TC-1 cell line was used to construct a mouse cervical cancer model, which comprises primary lung epithelial cells from C57BL/6 mice, immortalized with HPV16 E6 and E7 oncoproteins and transformed with the c-Ha-ras oncogene, donated by Prof. Yanbing Ma (Chinese Academy of Medical Sciences and Peking Union Medical College). TC-1 cells were cultured with RPMI 1640 (CORING, Corning, NY, USA) medium containing 10% fetal bovine serum (Thermo Fisher Scientific, MA, USA) in an incubator at 37 °C with 5% CO_2_. The C57BL/6 mice (female, 6–8 weeks of age) were obtained from the Central Animal Service Center, Institute of Medical Biology, Chinese Academy of Medical Sciences (IMBCAMS), and housed in specific pathogen-free (SPF) animal experimental barrier systems. All animal-related experiments were approved by the Laboratory Animal Ethics Committee of the IMBCAMS (approval number: DWSP2023060084).

### 2.9. IFN-γ ELISpot Assay

The C57BL/6 mice were randomly assigned to three groups, each containing four animals (*n* = 4). The groups were inoculated intramuscularly (i.m.) with 100 μL of either PBS, 20 μg of CpG ODN 1826, or E5E6E7pep11 (5 μg/peptide and 20 μg of ODN 1826 per mouse) as vaccine formulations. The mice were administered an immunization dose on days 0, 7, and 14. One week following the final immunization dose, the mice were euthanized, and their spleens were aseptically harvested. The spleens were then processed through a 70 µm cell strainer and subjected to Ficoll density gradient centrifugation to isolate single splenocytes. These cells were resuspended in a serum-free medium for subsequent cell counting.

The ELISpot assay was performed using the ELISpot Plus Kit: Mouse IFN-γ (Mabtech, Kista, Sweden) in accordance with the manufacturer’s instructions. In brief, 4 × 10^5^ cells were seeded into each well of a 96-well plate that had been pre-coated with an antibody to capture IFN-γ. The cells were incubated at 37 °C with 5% CO_2_ in the presence of stimulants for 24 h. Following incubation, the plate was washed, and the cells were sequentially incubated with R4-6A2-biotin antibody and streptavidin-ALP. The addition of BCIP/NBT-plus substrate facilitated the development of spots, which were subsequently enumerated using an ImmunoSpot^®^ Analyzer (Cellular Technology Limited, Shaker Heights, Cleveland, OH, USA). A total of 11 epitope peptides were combined to create a peptide pool (11p-pool) as a specific stimulant for cellular immune response with a final concentration of 2 μg/mL per peptide. Similarly, peptide pools containing E5 protein epitopes (E5-pool), E6 protein epitopes (E6-pool), and E7 protein epitopes (E7-pool) were formulated at equivalent single-peptide working concentrations. Serum-free medium was used as a negative stimulus control and PHA was used as a positive stimulus control. Three replicates were set up for each well.

### 2.10. In Vivo Tumor Prevention Studies

The C57BL/6 mice were randomly divided into 4 groups of 6 mice each (*n* = 6) and received intramuscular injections of 100 μL of E5E6E7pep11 (5 μg/peptide and 20 μg of CpG ODN 1826) or CTB-Epi11E567 (100 μg) at 7, 14, and 21 days prior to tumor challenge. Control groups were also established and administered with PBS and CpG ODN 1826 (20 μg) for comparison. The mice received a subcutaneous injection (s.c.) of 1 × 10^5^ TC-1 cells in the lower right dorsal area 7 days after the final vaccination. Tumor size was measured and recorded at three-day intervals post-inoculation. The endpoint of survival was determined when tumors reached the ethical limit (diameter = 15 mm), at which point the mice were euthanized.

### 2.11. In Vivo Tumor Treatment Study

The mice were grouped for treatment experiments in the same manner as in the prophylactic study. In contrast, the mice were pre-challenged with 1 × 10^5^ TC-1 cells on day 0 and given one dose of vaccine intramuscularly on days 3, 10, and 17 following tumor challenge. The dosage administered was consistent with that described in the tumor prevention study. Tumor measurements and survival endpoint determination were performed as previously described.

### 2.12. Statistical Analysis

Two-way ANOVA was used to statistically analyze the data from the ELISpot assay, and the log-rank (Mantel–Cox) test was used to compare and plot the survival curves, with a *p*-value of less than 0.05 considered statistically significant. All statistical analyses and graphing were carried out in GraphPad Prism 8.

## 3. Results

### 3.1. Epitope Prediction and Selection

Below, an integrated flowchart is used to illustrate the step-by-step process utilized for epitope prediction and selection. The brief results of our computational analyses are summarized in Figure 1.

In the immunoinformatics analysis, the binding affinities of HPV16 E5, E6, and E7 protein epitopes to the 70 global high-frequency HLA class I alleles, enumerated in Appendix A, were, respectively, predicted using the NetMHCPan 4.1 EL model. Epitopes with score values > 0.4 were retained, yielding a total of 80 E5 (Appendix A), 121 E6 (Appendix A), and 61 E7 epitopes (Appendix A).

To enhance the cross-reactivity of epitopes among different HPV16 variants, we performed a conservation analysis of epitopes that scored higher than the cut-off value and removed epitopes that were more than 95% conserved among the 68 reference variants, including 16 E5 (Appendix A), 35 E6 (Appendix A), and 22 E7 epitopes (Appendix A). To facilitate subsequent functional validation of epitopes in a mouse model, we conducted CTL epitope prediction based on the H-2b alleles. Upon setting a threshold with a total score > 0, we identified 45 E5, 22 E6, and 19 E7 H-2b-restricted CTL epitopes, as shown in Appendix A. In the final selection process, conserved CTL epitopes restricted by both human and murine MHC molecules were retained as candidates for vaccine construct development, including 3 E5, 4 E6, and 4 E7 epitopes. The locations of the predicted CTL/CD8^+^ T cell epitopes in the E5, E6, and E7 proteins were mapped and are presented in Figure 2. In addition, the HLA-I alleles presenting these epitopes are also annotated.

### 3.2. Characterization of CTL Epitopes for Vaccine Development

The results of the population coverage analysis, as shown in Table 1 and Appendix A, indicate that the combination of the 11 candidate epitopes can cover 98.42% of the global population. Notably, epitope E7p9 demonstrates the highest coverage rate, reaching 70.50%, thus suggesting that it possesses extensive potential applicability. We predicted the immunogenicity and antigenicity of 14 candidate epitopes, and the results, as presented in Table 1, indicate that epitopes E5p2, E5p3, E6p4, E6p5, E7p9, and E7p10 exhibit favorable immunogenicity (scores > 0). Moreover, E6p4, E7p9, E7p10, and E7p11 demonstrate pronounced antigenicity (scores > 0.4). In the allergenicity evaluation, the AllerTOP v.2 analysis classified only E6p5, E7p10, and E7p11 as potential allergens. With regard to toxicity, the prediction results of the ToxinPred sever revealed that all 11 candidate epitopes were non-toxic.

### 3.3. Evaluation of Antigen-Specific IFN-γ Response

To evaluate the ability of the candidate epitopes to induce specific cellular immune responses in vivo, immunogenicity studies were conducted in C57BL/6 mice. Following three doses of immunization at one-week intervals with a mixture of epitope peptides formulated with the CpG ODN 1826 adjuvant, E5E6E7pep11, the ability of splenocytes from the mice to secrete antigen-specific IFN-γ upon stimulation with peptide pools was assessed using the ELISpot assay. As shown in Figure 3, E5E6E7pep11-immunized mice induced higher levels of antigen-specific IFN-γ secretion compared to the other two groups; CpG injection alone did not induce IFN-γ secretion from CD8+ T cells. In addition, the E5 epitope was identified as the probable dominant epitope for triggering specific immune responses.

### 3.4. Construction and Expression of Recombinant Multi-Epitope Protein

Following the confirmation of the immunogenic efficacy of the mixed peptide vaccine, we proceeded to develop a recombinant protein vaccine comprising 11 epitopes for subsequent in vivo anti-tumor efficacy studies. Based on the amino acid sequence, we predicted the physicochemical properties of the vaccine expression product CTB-Epi11E567, which contains a total of 302 amino acids and possesses a molecular weight (MW) of 34.05 kDa, a theoretical isoelectric point (pI) of 8.27, an aliphatic index of 91.59, an instability index of 34.80, and a grand average of hydropathicity (GRAVY) of −0.007. These results indicate that the long peptide expressed by our therapeutic vaccine was hydrophilic and stable.

### 3.5. Prophylactic Efficacy of the Multi-Epitope Vaccine

The prophylactic anti-tumor effects of E5E6E7pep11 and CTB-Epi11E567 were further evaluated using a syngeneic model grafted with TC-1. As in the immunization program shown in Figure 4A, the C57BL/6 mice were immunized with three separate doses at one-week intervals prior to subcutaneous challenge with TC-1 cells. The survival status of the mice was continuously recorded for a period of two months following tumor challenge, and the survival curves for each group were plotted. Mice in the PBS placebo control group and the CpG adjuvant-only control group exhibited continuous tumor growth, and all reached ethical endpoint criteria within 40 days post-tumorigenesis. In contrast, vaccination with the E5E6E7pep11 and CTB-Epi11E567 vaccines demonstrated a complete prophylactic anti-tumor effect. The survival rate in mice was 100%, and the state of being tumor-free was maintained until the end of the observation period, which was day 80 post-challenge (Figure 4B).

### 3.6. Therapeutic Efficacy of the Multi-Epitope Vaccine

The therapeutic potential of E5E6E7pep11 and CTB-Epi11E567 in treating established tumors was also assessed. The mice were subjected to a subcutaneous challenge of 1 × 10^5^ TC-1 cells on day 0, followed by 1 dose of vaccination on days 3, 10, and 17. The tumor volume and survival status of the mice were continuously monitored and recorded. Mice whose tumor volumes met the ethical endpoint criteria were euthanized. In cases where no tumor development occurred or tumor progression was slow, the mice were uniformly euthanized upon the conclusion of the observation period, which was day 80 post-tumor challenge (Figure 5A). In the above experiments, the CpG adjuvant control group displayed complete non-immunogenicity and no antitumor efficacy as expected; therefore, this group was omitted from the current assay. Instead, a mixture of peptide pep11 group without adjuvants was included to investigate the contribution of adjuvants to the therapeutic efficacy of the mixed epitope peptide. In comparison with the PBS control group, mice treated with E5E6E7pep11 and CTB-Epi11E567 vaccines exhibited significantly slower tumor growth, with approximately 67% (4/6) and 50% (3/6) of the mice remaining tumor free up to 80 days post tumor challenge, respectively. Furthermore, despite the favorable anti-tumor therapeutic efficacy evinced by E5E6E7pep11, vaccination with the adjuvant-free mixed peptides (pep11) failed to achieve the desired therapeutic effect, with no significant survival benefit observed in this group as compared to the PBS control group (Figure 5B,C).

## 4. Discussion

Although a variety of therapeutic HPV vaccines are already under active development, such as VGX3100, a DNA vaccine targeting the HPV16 E6/E7 fusion protein, there is currently no successful vaccine on the market to eliminate established HPV infections [5]. Relying on traditional laboratory strategies, vaccine development is generally a time-consuming and costly endeavor [22]. Rapid advances in biotechnology and immunoinformatic technology have made it possible to rapidly identify potential antigenic epitopes by means of huge amounts of immune library data and machine learning-based information tools [23]. In the current study, we harnessed the predictive capacity of the NetMHCPan 4.1 EL method for in silico prediction to expedite the identification and screening of antigenic epitopes [24,25]. Fortified by a substantial corpus of experimental data, the algorithm was proven to be capable of pinpointing peptide fragments that are potential candidates for binding to specific MHC class molecules, thus potentially triggering T-cell responses. We ascertained a total of 16 E5 epitopes, 35 E6 epitopes, and 22 E7 epitopes, and have corroborated the immunogenic potential of 11 CTL epitopes in B6 mice.

The epitope recognition of CTLs is restricted by specific human leukocyte antigen (HLA) molecules, which are highly polymorphic in the general population. Therefore, screening for antigenic peptides that bind to HLA in the broader population is critical for the development of therapeutic vaccines to enable mass population vaccination [26,27,28]. The overall population coverage of the candidate epitope combinations that we ultimately obtained reached 98.42%, indicating that vaccines constructed based on these epitopes are suitable for populations with diverse genetic backgrounds. Specifically, individual epitopes such as E5p2, E6p4, E6p5, E6p6, E7p8, and E7p9 can cover over 30% of the general population; notably, the E7p9 epitope was able to sufficiently bind with 20 HLA-I class molecules, implying a broader applicability of these epitope vaccines across different ethnic groups.

Current therapeutic strategies for HPV16 are focused on E6 and E7, two proteins closely associated with cervical carcinogenesis [9,10,11,29]. In this study, we included the E5 oncogenic protein as one of the target antigens and used three CTL candidate epitopes located in the conserved region of the E5 protein to construct a multi-epitope vaccine. Some studies have suggested that the E5 protein could be a potential target for cervical cancer therapy [30,31]. Although frequently deleted in the advanced stages of cervical cancer, the E5 protein indeed participates in multiple signaling pathways regulating malignant transformation [12,32,33,34,35]. E5 is widely expressed on the surface of cervical epithelial cells in the early stages of cervical lesions; therefore, the targeting of immune epitopes of the E5 protein could represent a promising therapeutic strategy [36]. Yi-Fang Chen et al. identified VCLLIRPLL_25–33_ as a restricted CTL epitope in Db-C57BL/6 mice and verified in a mouse model that inoculation of this epitope peptide in combination with CpG ODN 1826 induced HPV16-associated tumor elimination [30]. The design of the multi-epitope vaccine reflects a comprehensive strategy aimed at enhancing the immune response by concurrently targeting multiple oncogenic proteins of HPV16. Compared to vaccines that target a limited number of antigens, multi-epitope vaccines theoretically offer more comprehensive protection by targeting a broader spectrum of antigens. For example, in a previous study, Liao S et al. demonstrated that the incorporation of the E5 peptide alongside E6 and E7 peptides significantly enhanced tumor protection, surpassing the protective effects observed with either E5 alone or E6 + E7 alone [37]. In the validation assay of epitope immunogenicity, we stimulated splenocytes using different peptide pools in vitro with the aim of carrying out a dominant epitope analysis, and the results showed that the E5 epitope peptide combination appeared to be the main reason for the cells being induced to secrete antigen-specific IFN-γ, which suggested that the E5 antigenic epitope may play a role in anti-tumor immunity. However, since the mouse tumor model used in this study was constructed based on the TC-1 cell line, which does not express the E5 oncoprotein on its surface, the demonstrated anti-tumor efficacy might be derived to a greater degree from the E6 and E7 epitopes. Several previous studies have demonstrated the good immunogenicity and antitumor protective effects of some of our candidate E6 and E7 epitope peptides, similar to our results on the TC-1 tumor model [38,39,40,41]. For instance, Jemon et al. developed a heterologous virus-like particle vaccine containing the EVYDFAFRDL_48–57_ epitope peptide from the HPV16 E6 protein and the helper T-cell epitope PADRE, resulting in a significant delay in tumor growth and improved survival rates in a TC-1 mouse tumor model [40]. In another study, He X et al. combined the E7HHH_49–57_ epitope peptide with CoPoP liposomes to create a peptide-liposome vaccine. This formulation successfully triggered an effective CD8+ T-cell response in mice and demonstrated efficacy in eradicating or reversing TC-1 tumor growth [41]. This evidence presented indicated that while the E6 and E7 epitopes did not substantially stimulate antigen-specific IFN-γ secretion in immunogenicity assessment assays, this does not rule out the immunogenic potential of these epitopes. This observation could be due to the immune system’s preferential recognition of one or more E5 epitope peptides, leading to competitive inhibition in recognizing the other epitopes. Assessment of vaccine protective efficacy on a TC-1 tumor model expressing E5 protein is very valuable for gaining deeper insights into the contribution of E5, E6, and E7 epitopes in antitumor immunity, which is worth further exploring in future studies.

Although the results of some studies have substantiated the therapeutic anti-tumor efficacy of minimal epitope peptides, there is also evidence suggesting that synthetic long peptide vaccines may elicit more robust antigen-specific CD8^+^ T cell responses and potentiate greater tumor treatment effects compared to short peptide vaccines. This could be attributed to the additional steps of antigen uptake, processing, and presentation by antigen-presenting cells (APCs) that long peptides undergo in the body [42,43,44]. The immune efficacy of the recombinant protein vaccine in the form of a synthetic long peptide and a mixed minimal epitope peptide vaccine was assessed simultaneously in our study, and we observed that the long peptide vaccine, CTB-Epi11E567, appeared to possess an advantage in terms of therapeutic anti-tumor efficacy. However, this result could be ascribed to disparities in the vaccine dosage. The authors of future studies may further explore the immunogenicity of the vaccine across a broader range of doses. Overall, both vaccine modalities demonstrated significant prophylactic and therapeutic anti-tumor effects in mice. Additionally, the significance of adjuvants in peptide-based vaccine formulations has been underscored in previous studies [43,45,46,47]. In this study, to augment the immunogenicity of the vaccine, we incorporated the adjuvant CpG ODN 1826, an agonist of pattern-recognition receptor nine (TLR9) widely used as a vaccine adjuvant, into the peptide vaccine. We discerned that the CpG adjuvant played a crucial role when combined with the short peptide form of the vaccine owing to the fact that the inoculation with epitope peptides alone did not engender effective anti-tumor protection, aligning with the results of previous studies [48]. The cholera toxin B subunit (CTB) has long been recognized as a potent mucosal adjuvant, and evidence from a recent study indicates its promise as a systemic adjuvant [49]. In this study, CTB was used as an intramolecular adjuvant to prepare recombinant protein vaccines to facilitate broader in vivo immune stimulation. In addition, we found that N-terminal fusion CTB conferred better stability and solubility of the multi-epitope peptide as evidenced by the difference in the physicochemical properties of CTB-Epi11E567 and Epi11E567, which was more conducive to protein expression, purification, and preservation.

In this study, we present the available epitope combinations and validate the immune effects of these epitopes in mice. However, the differences between mouse and human MHCs led to some limitations in the epitope screening process, as some conserved CTL epitopes with high scores were not selected for further experimental validation in mice. The construction of H-2 humanized mouse models may facilitate further in vivo evaluations of more epitopes in the future. Additionally, both forms of the multi-epitope vaccines achieved a 100% anti-tumor protective effect in prophylactic anti-tumor trials; however, there was a lack of evaluation of long-term immune protection and protective efficacy following re-challenge with the tumor. The detection of immune memory cells will contribute to a more comprehensive evaluation of the vaccines. As helper CD4 T cells play a crucial role in enhancing the proliferation of CTL clones and guiding their transformation into effector and memory cells, incorporating Th epitopes may direct the vaccine-induced anti-tumor response towards sustainable and comprehensive regulation [50,51,52].

## 5. Conclusions

In this study, we employed immunoinformatics approaches to identify 11 conserved cytotoxic T lymphocyte (CTL) epitopes targeting the HPV16 oncoproteins. Subsequently, we developed peptide vaccines and recombinant protein vaccines incorporating these epitopes. Initial assessment of the immunogenicity of these epitopes and the anti-tumor efficacy of the vaccines was conducted using a murine tumor model. Our findings present a promising selection of candidate epitopes and introduce a viable approach for the rapid development of therapeutic multi-epitope vaccines against HPV16, with the potential for widespread population coverage.

## Figures and Tables

**Figure 1 vaccines-12-00392-f001:**
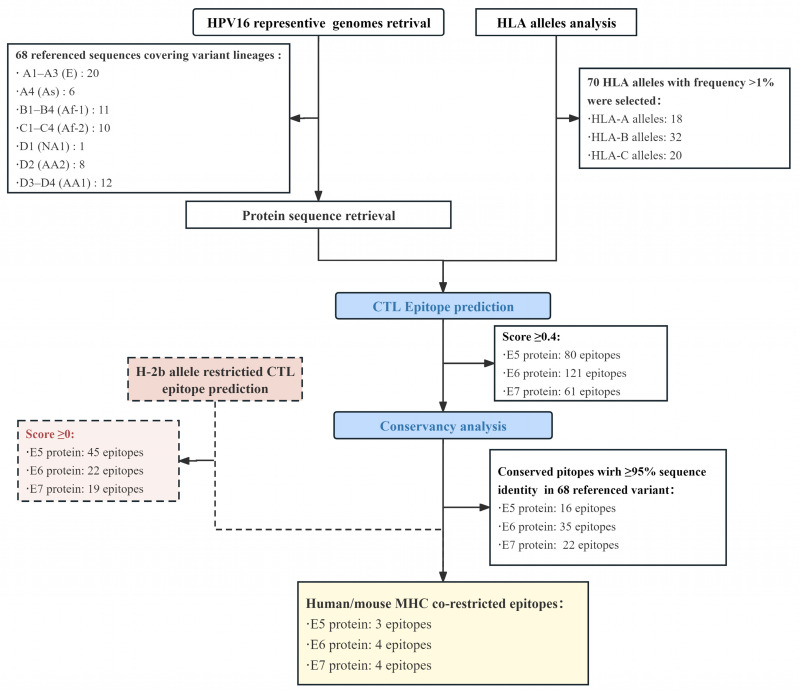
Flowchart and brief results of epitope prediction and screening in the present study.

**Figure 2 vaccines-12-00392-f002:**
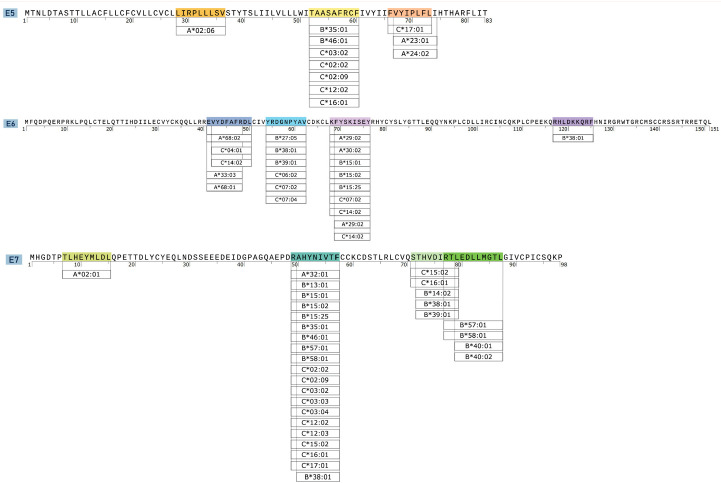
CTL/CD8^+^ candidate epitope location maps plotted for the E5, E6, and E7 proteins.

**Figure 3 vaccines-12-00392-f003:**
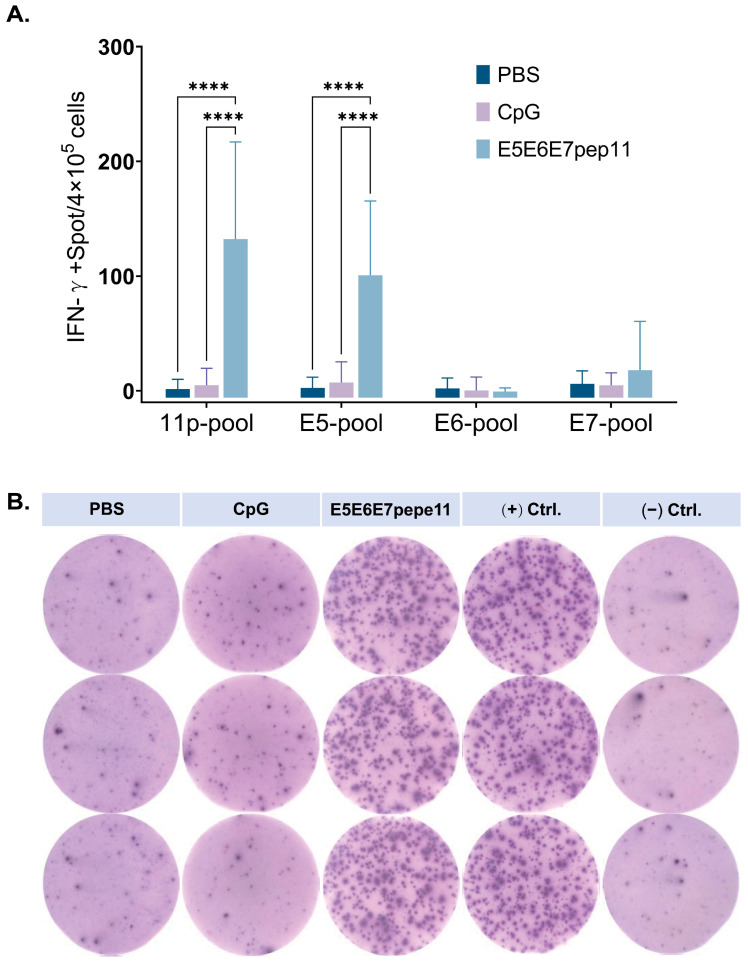
Induction of antigen-specific IFN-γ response in C57BL/6 mice vaccinated with E5E6E7pep11. C57BL/6 mice (*n* = 4) were inoculated intramuscularly (i.m.) with 100 μL of either PBS, CpG ODN 1826, or E5E6E7pep11 on days 0, 7, and 14. Splenic lymphocytes were collected 1 week after the second immunization process and then stimulated in vitro with the 11p-pool, E5-pool, E6-pool, and E7-pool. (**A**) The number of IFN-γ spot-forming cells (SFCs) detected using ELISpot was subtracted from the SFCs stimulated by serum-free medium as a negative control. (**B**) Representative ELISpot images, using 11p-pool as a specific stimulus, PMA as a non-specific stimulus (positive control), and serum-free medium as a negative stimulus (negative control). **** *p* < 0.0001.

**Figure 4 vaccines-12-00392-f004:**
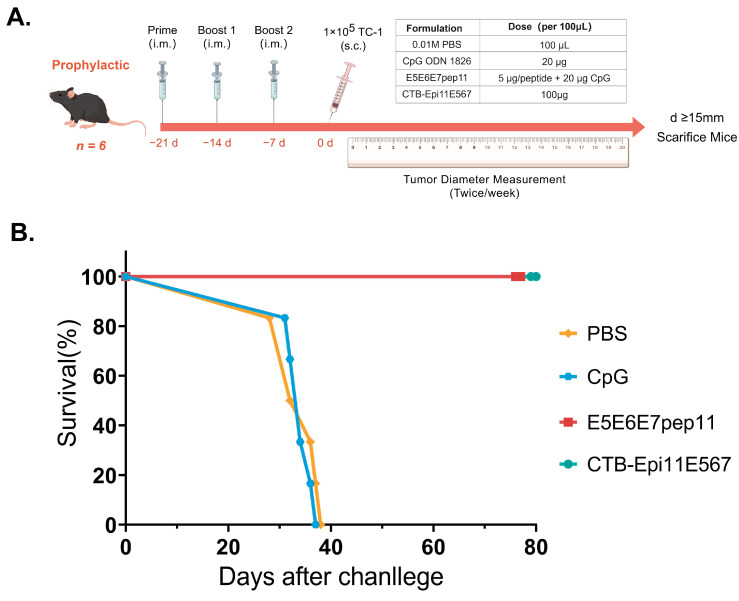
Prophylactic anti-tumor immunity induced by multi-epitope vaccines. (**A**) The C57BL/6 mice (*n* = 6) were intramuscularly vaccinated (i.m.) with 100 μL of PBS, CpG ODN 1826 (20 μg), E5E6E7pep11 (5 μg/peptide and 20 μg CpG ODN 1826), or CTB-Epi11E567 (100 μg) at 7, 14, and 21 days prior to the 1 × 10^5^ TC-1 cell challenge. (**B**) Percentage survival of each group of mice within 80 days of tumor challenge. The mice were euthanized when their tumor diameter reached ≥ 15 mm.

**Figure 5 vaccines-12-00392-f005:**
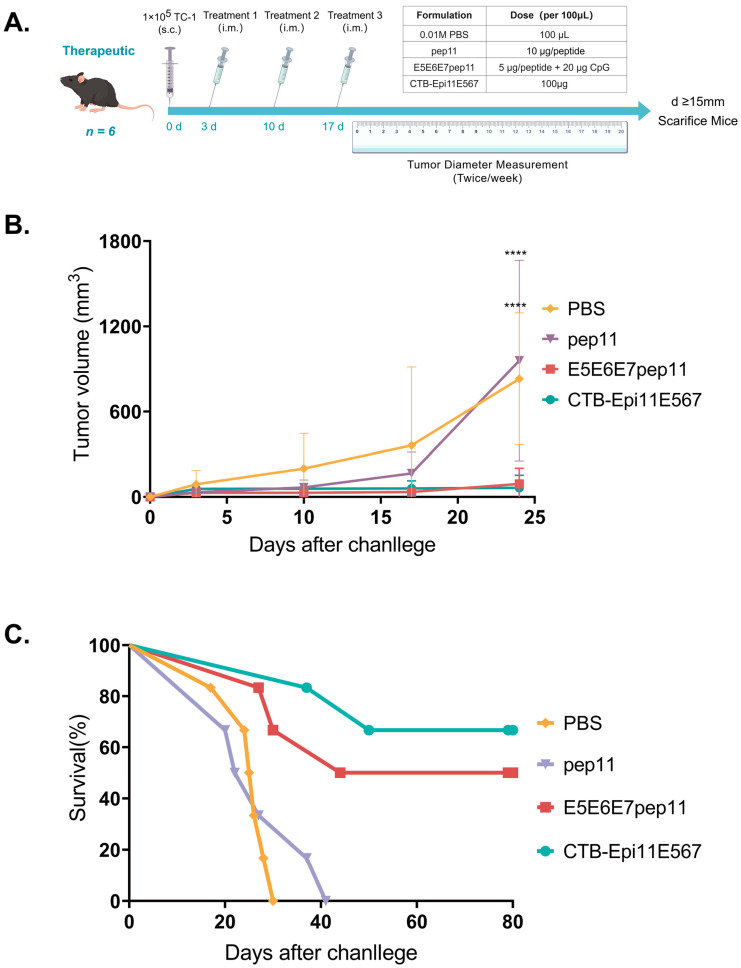
Therapeutic anti-tumor immune effects induced by multi-epitope vaccines. (**A**) The C57BL/6 mice (n = 6) were intramuscularly vaccinated (i.m.) with 100 μL of PBS, pep11 (5 μg/peptide), E5E6E7pep11 (5 μg/peptide and 20 μg CpG ODN 1826), or CTB-Epi11E567 (100 μg) at 3, 10, and 17 days post-challenge of 1 × 10^5^ TC-1 cells. (**B**) Tumor volume was measured biweekly, and the average tumor sizes for each group of mice were computed to graph the kinetics of tumor growth. (**C**) Percentage survival of each group of mice within 80 days of tumor challenge. **** *p* < 0.0001.

**Table 1 vaccines-12-00392-t001:** Characterization of conserved CTL candidate epitopes for vaccine construction.

Protein	Epitope	Sequence	Position	Length	Coverage	Immunogenicity	Antigency	Allergenicity	Toxicity
E5	E5p1	LIRPLLLSV	28–36	9	1.95%	−0.12	0.33	Non-Allergen	Non-Toxin
	E5p2	TAASAFRCF	52–60	9	30.81%	0.01	0.26	Non-Allergen	Non-Toxin
	E5p3	FVYIPLFLI	66–73	9	28.62%	0.2	0.39	Non-Allergen	Non-Toxin
E6	E6p4	EVYDFAFRDL	41–50	10	32.66%	0.34	1.46	Non-Allergen	Non-Toxin
	E6p5	YRDGNPYAV	54–62	9	44.37%	0.04	0.39	Allergen	Non-Toxin
	E6p6	KFYSKISEY	68–76	9	37.65%	−0.33	0.23	Non-Allergen	Non-Toxin
	E6p7	RHLDKKQRF	117–125	9	3.23%	−0.46	0.35	Non-Allergen	Non-Toxin
E7	E7p8	TLHEYMLDL	7–15	9	39.08%	−0.05	0.33	Non-Allergen	Non-Toxin
	E7p9	RAHYNIVTF	49–57	9	70.50%	0.18	0.59	Non-Allergen	Non-Toxin
	E7p10	STHVDIRTL	71–79	9	16.98%	0.27	0.58	Allergen	Non-Toxin
	E7p11	RTLEDLLMGTL	77–87	11	17.96%	−0.02	0.64	Allergen	Non-Toxin

## Data Availability

The original contributions presented in this study are included in the article and Appendix A, further inquiries can be directed to the first authors and corresponding authors.

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
