# Peer review of "Immunoinformatics Design and In Vivo Immunogenicity Evaluation of a Conserved CTL Multi-Epitope Vaccine Targeting HPV16 E5, E6, and E7 Proteins"

_vaccines, 2024, doi:10.3390/vaccines12040392_

Round 1

Reviewer 1 Report

Comments and Suggestions for Authors

adding more recent references is beneficial

Comments on the Quality of English Language

minor English corrections are needed

Author Response

Response to Reviewer 1 Comments

1. Summary

Thank you very much for taking the time to review this manuscript. Please find the detailed responses below and the corresponding corrections highlighted changes in the re-submitted files.

2. Questions for General Evaluation

Reviewer’s Evaluation

Response and Revisions

Does the introduction provide sufficient background and include all relevant references?

Yes

Are all the cited references relevant to the research?

Yes

Is the research design appropriate?

Yes

Are the methods adequately described?

Yes

Are the results clearly presented?

Yes

Are the conclusions supported by the results?

Yes

3. Point-by-point response to Comments and Suggestions for Authors

Comments: Adding more recent references is beneficial.

Response: We sincerely appreciate the valuable comments. We have checked the literature carefully and added nine references into the Discussion part in the revised manuscript, five of which were published within the last five years..

4. Response to Comments on the Quality of English Language

Point: Minor English corrections are needed.

Response: Thank you for your kind reminder. We have now worked on both language and readability and have also involved native English speakers for language corrections through MDPI (engilish-78884). The edited English have been marked in red.

5. Additional clarifications

In response to the comments from all peer reviewers, we have incorporated additional content supplements and modifications into the manuscript, which are highlighted within the text.

Reviewer 2 Report

Comments and Suggestions for Authors

In this paper, the authors have re-evaluated an established approach to vaccine development on the basis of peptides selected for their immunogenicity using recently-developed software and established validation approaches. The authors have addresses the issues of variations in HPV oncoprotein sequences and MHC alleles worldwide and have derives conserved MHC class I epitopes from the HPV E5, E6 and E7 proteins that they have incorporated into a fusion protein vaccine. This seems to have efficacy in protection from and eradication of a mouse HPV-driven tumour. Overall, this work represents a new variant on an established approach. 

Comments on the Quality of English Language

There are a number of English spelling and grammatical errors (e.g. the word "conservancy" does not exist and should be replaced by conservation). More importantly, there are typographical errors in many of the Figures that need to be corrected.

Author Response

Response to Reviewer 2 Comments

1. Summary

Thank you very much for taking the time to review this manuscript. Please find the detailed responses below and the corresponding revisions changes in the re-submitted files.

2. Questions for General Evaluation

Reviewer’s Evaluation

Response and Revisions

Does the introduction provide sufficient background and include all relevant references?

Yes

Are all the cited references relevant to the research?

Yes

Is the research design appropriate?

Yes

Are the methods adequately described?

Yes

Are the results clearly presented?

Yes

Are the conclusions supported by the results?

Yes

3. Point-by-point response to Comments and Suggestions for Authors

Comments: In this paper, the authors have re-evaluated an established approach to vaccine development on the basis of peptides selected for their immunogenicity using recently-developed software and established validation approaches. The authors have addresses the issues of variations in HPV oncoprotein sequences and MHC alleles worldwide and have derives conserved MHC class I epitopes from the HPV E5, E6 and E7 proteins that they have incorporated into a fusion protein vaccine. This seems to have efficacy in protection from and eradication of a mouse HPV-driven tumour. Overall, this work represents a new variant on an established approach.

Response: We feel great thanks for your professional review on our article and kind recognition of our work.

4. Response to Comments on the Quality of English Language

Point: There are a number of English spelling and grammatical errors (e.g. the word "conservancy" does not exist and should be replaced by conservation). More importantly, there are typographical errors in many of the Figures that need to be corrected.

Response: We apologize for the poor language and mistakes of our manuscript. We have now worked on both language and readability and have also involved native English speakers for language corrections through MDPI (engilish-78884). The edited English have been marked in red. We really hope that the flow and language level have been substantially improved. In addition, we re-corrected the figures and tables throughout the manuscript. Thank you for your reminder.

5. Additional clarifications

In response to the comments from all peer reviewers, we have incorporated additional content supplements and modifications into the manuscript, which are highlighted within the text.

Reviewer 3 Report

Comments and Suggestions for Authors

The paper you've described is excellent. It presents a comprehensive and innovative approach to the development of a multi-epitope vaccine targeting HPV16, which is a significant causative agent of cervical cancer worldwide. The use of immunoinformatics to predict and characterize CTL epitopes across the E5, E6, and E7 proteins of HPV16, and the subsequent validation of these epitopes' immunogenicity and anti-tumor efficacy in a mouse model, are both cutting-edge strategies. The findings that both the peptide and recombinant protein vaccines could induce strong immune responses and provide complete protection against tumor growth in mice are particularly promising. This work not only advances our understanding of HPV immunotherapy but also lays a solid foundation for future vaccine development against HPV-associated cancers. Overall, the paper is well-executed, with significant implications for the field of cancer immunotherapy.

Comments on the Quality of English Language

Minor editing of English language required

Author Response

Response to Reviewer 3 Comments

1. Summary

Thank you very much for taking the time to review this manuscript. Please find the detailed responses below and the corresponding revisions changes in the re-submitted files.

2. Questions for General Evaluation

Reviewer’s Evaluation

Response and Revisions

Does the introduction provide sufficient background and include all relevant references?

Yes

Are all the cited references relevant to the research?

Yes

Is the research design appropriate?

Yes

Are the methods adequately described?

Yes

Are the results clearly presented?

Yes

Are the conclusions supported by the results?

Yes

3. Point-by-point response to Comments and Suggestions for Authors

Comments: The paper you've described is excellent. It presents a comprehensive and innovative approach to the development of a multi-epitope vaccine targeting HPV16, which is a significant causative agent of cervical cancer worldwide. The use of immunoinformatics to predict and characterize CTL epitopes across the E5, E6, and E7 proteins of HPV16, and the subsequent validation of these epitopes' immunogenicity and anti-tumor efficacy in a mouse model, are both cutting-edge strategies. The findings that both the peptide and recombinant protein vaccines could induce strong immune responses and provide complete protection against tumor growth in mice are particularly promising. This work not only advances our understanding of HPV immunotherapy but also lays a solid foundation for future vaccine development against HPV-associated cancers. Overall, the paper is well-executed, with significant implications for the field of cancer immunotherapy.

Response : We feel great thanks for your professional review on our article and kind recognition of our work.

4. Response to Comments on the Quality of English Language

Point: Minor editing of English language required

Response: Thank you for your kind reminder. We have now worked on both language and readability and have also involved native English speakers for language corrections through MDPI (engilish-78884). The edited English have been marked in red.

5. Additional clarifications

In response to the comments from all peer reviewers, we have incorporated additional content supplements and modifications into the manuscript, which are highlighted within the text.

Reviewer 4 Report

Comments and Suggestions for Authors

Vaccines based on cytotoxic T-lymphocyte (CTL) epitopes are a promising strategy for the elimination of pre-existing HPV infections and treating cervical cancer. The current study screened and tested a multi-epitope vaccine (E5E6E7pep11) based on three E5, four E6 and four E7 epitopes in vitro and in vivo. Anti-tumor effects of E5E6E7pep11 and the recombinant protein vaccine CTB- Epi11E567 were evaluated in the TC-1 mouse tumor model. Complete protection was found by immunization with E5E6E7pep11 and CTB-Epi11E567 TC-1 tumor growth in mice. These two vaccines also inhibited tumor growth and prolonged mouse survival. While including epitopes from multiple proteins is a useful way to increase the efficacy of the anti-tumor effect, the experimental design needs to be improved and strengthened.

The TC-1 cells express HPV16 E6/E7 not E5. The rationale for using TC-1 in vivo model to test E5 epitopes needs to be clarified. It’s interesting that the highest responses were found to E5 epitopes in vitro. The correlation between this response intro and in vivo would benefit from a cell line expressing all three proteins (e.g TC-1 expressing E5). It would be interesting to determine whether the multi-epitope E5E6E7pep11 vaccine provides better therapeutic effect than vaccines targeting either E6 or E7 in previous studies. Interestingly even with minimal response to E6 and E7 epitopes in the in vitro assay, you detected complete protection and decent therapeutic effect. Do you think the suboptimal therapeutic effect resulted from the long immune responses to E6 and E7 by the vaccine?  

Table 1: please add the position of the epitopes in the proteins.

Please offer justification for using different doses as well as different control groups for comparison

in in vitro study

PBS, CpG ODN 1826 (20μg), E5E6E7pep11 (5 μg/peptide and 20 μg CpG ODN 1826) or CTB-Epi11E567 (100 μg)

And in vivo

PBS, pep11 (5 μg/peptide), E5E6E7pep11 (5 μg/peptide and 20 μg CpG ODN 1826) or CTB-Epi11E567 (100 μg)

Line 339 “67% (4/6) and 50% (3/6) of the mice remaining tumor-free up to 80 days post-tumor challenge.”  You treated these animals three times for therapy. Would additional treatments increase the tumor-free survival in these mice?

Comments on the Quality of English Language

Proofreading by a native English editor would help to increase the readership. 

Author Response

Response to Reviewer 4 Comments

1. Summary

Thank you very much for taking the time to review this manuscript. Please find the detailed responses below and the corresponding corrections highlighted changes in the re-submitted files.

2. Questions for General Evaluation

Reviewer’s Evaluation

Response and Revisions

Does the introduction provide sufficient background and include all relevant references?

Can be improved

Further additions have been made.

Are all the cited references relevant to the research?

-

Corrections and refinements have been made

Is the research design appropriate?

Can be improved

Discussion of limitations has been added

Are the methods adequately described?

Can be improved

Corrections and refinements have been made

Are the results clearly presented?

Can be improved

Further additions have been made.

Are the conclusions supported by the results?

Can be improved

3. Point-by-point response to Comments and Suggestions for Authors

Comments 1: Vaccines based on cytotoxic T-lymphocyte (CTL) epitopes are a promising strategy for the elimination of pre-existing HPV infections and treating cervical cancer. The current study screened and tested a multi-epitope vaccine (E5E6E7pep11) based on three E5, four E6 and four E7 epitopes in vitro and in vivo. Anti-tumor effects of E5E6E7pep11 and the recombinant protein vaccine CTB- Epi11E567 were evaluated in the TC-1 mouse tumor model. Complete protection was found by immunization with E5E6E7pep11 and CTB-Epi11E567 TC-1 tumor growth in mice. These two vaccines also inhibited tumor growth and prolonged mouse survival. While including epitopes from multiple proteins is a useful way to increase the efficacy of the anti-tumor effect, the experimental design needs to be improved and strengthened.

Response 1: We feel great thanks for your professional review work on our article. We regret that there are still imperfections in our experimental design, and we will respond to your concerns point by point in the following comments.

Comments 2: The TC-1 cells express HPV16 E6/E7 not E5. The rationale for using TC-1 in vivo model to test E5 epitopes needs to be clarified. It’s interesting that the highest responses were found to E5 epitopes in vitro. The correlation between this response intro and in vivo would benefit from a cell line expressing all three proteins (e.g TC-1 expressing E5).

Response 2: Thank you for pointing this out. We fully agree with your comments,and indeed we are making further efforts in this regard. In this study, we utilized TC-1 cells to construct a tumor model. These cells are syngeneic with C57BL/6 mice and can continuously express HPV16 E6 and E7 proteins, displaying similar biological behavior to cervical cancer cells. Therefore, they are the most widely used tumor model for evaluating cervical cancer-related vaccines and therapeutic drugs currently. Utilizing the TC-1 tumor model will facilitate comparison of our results with past related research. As you have pointed out, due to the inclusion of the E5 epitope in the vaccine, research based on the TC-1 tumor model makes it challenging to demonstrate the vaccine's anti-tumor effects targeting the E5 protein. Further validation of the protective effect on E5-expressing TC-1 tumour models would help to more comprehensively evaluate the efficacy of vaccines containing E5 antigenic epitopes, and we are actively working on this, but we have not succeeded and are keeping on it. We have added a discussion of the limitations of the study (Supplemented in the Discussion section of the manuscript- - page 12, Phase 3, line 437–440, highlighted).

Comments 3: It would be interesting to determine whether the multi-epitope E5E6E7pep11 vaccine provides better therapeutic effect than vaccines targeting either E6 or E7 in previous studies. Interestingly even with minimal response to E6 and E7 epitopes in the in vitro assay, you detected complete protection and decent therapeutic effect. Do you think the suboptimal therapeutic effect resulted from the long immune responses to E6 and E7 by the vaccine?

Response 3: We couldn't agree more with your insights, and Your suggestion provides a direction for our next research. As stated in the previous response, since the tumor model expressing E5 is still in the process of being constructed, we did not conduct further comparisons of the therapeutic efficacy of E5E6E7pep11 versus other vaccines that target only the E6 or E7 antigens. We acknowledge a prior study that compared the therapeutic effectiveness of vaccines incorporating E5, E6, and E7 epitopes against vaccines containing solely E5 or E6+E7, and this has been included in the Discussion section. (Supplementary to the Discussion section of the manuscript - page 12, Phase 3, line 408-415, highlighted). Although we found in this study that the E5 epitope was critical for antigen-specific IFN-γ induction in our in vitro stimulation assay, effective immunogenicity and protective results for E6 and E7 epitope peptides overlapping with our candidate peptides have also been reported in previous studies. Therefore, we believe that it was possible that one or more of the E5 epitopes had higher immunogenicity and could be more preferentially recognized and activated by the immune system, resulting in the immunogenicity of the other epitopes being suppressed or masked and thus not immunodominant, and therefore these epitope peptide pools showed negative results in the in vitro IFN-γ stimulation assay, rather than negating the immunogenic potential of the E6 and E7 epitopes. The protective effect of our vaccines in the TC-1 tumour model further validates our conjecture, so we consider the results in this section to be of value (Supplementary to the Discussion section of the manuscript - page 12, Phase 3, line 423-437, highlighted). We believe that including more antigenic targets can broaden and strengthen the T cell immune response, thereby enhancing the therapeutic effect. However, further research is necessary to fully investigate the mechanisms underlying tumor immunity.

Comments 4: Table 1: please add the position of the epitopes in the proteins.

Response 4: We think this is an excellent suggestion. We have added a Position column in Table 1 to show the position of each epitope in the corresponding protein.

Comments 5: Please offer justification for using different doses as well as different control groups for comparison in in vitro study PBS, CpG ODN 1826 (20μg), E5E6E7pep11 (5 μg/peptide and 20 μg CpG ODN 1826) or CTB-Epi11E567 (100 μg) And in vivo PBS, pep11 (5 μg/peptide), E5E6E7pep11 (5 μg/peptide and 20 μg CpG ODN 1826) or CTB-Epi11E567 (100 μg).

Response 5: Thank you for pointing this out.

The reasons for having different control groups in different experiments are explained as follows:

l  Fistly, as we describe in the manucript, the comparison of immunogenicity vaccination and in vitro studies of PBS, CpG ODN1826, and E5E6E7pep11 was conducted to assess the ability of the candidate epitopes to induce specific cellular immune responses in vivo.

l  And,following the confirmation via the ELISpot assay that the epitope peptides could induce cellular immunity in healthy mice, we further developed the recombinant protein vaccine CTB-Epi11E567 and subsequently assessed the anti-tumor effects of both E5E6E7pep11 and CTB-EpiE567 in further studies.

l  In order to minimize use of mice, after we confirmed that the immune response induced by the vaccine was unrelated to the CpG ODN1826 adjuvant alone in the immunogenicity and prophylactic inoculation experiments, the adjuvant group was omitted in the subsequent comparisons, and replaced it with a non-adjuvanted mixture of peptides, pep11 group.

l  By comparing the non-adjuvanted antigen control group pep11 and the adjuvanted vaccine treatment group E5E6E7pep11 containing the same antigens, we evaluated the importance of adjuvants in enhancing the vaccine's immunoprotective properties. We have made the necessary additions to the manuscript. (Supplementary to the Result section of the manuscript - page 10, Phase 1, line 347–352, highlighted).

In terms of dose confirmation, we referred to preclinical trial data from the HPV16 recombinant protein vaccine TA-CIN(32μg~200μg/mice), synthetic peptide vaccine Pepcan (50μg/peptide, 3 peptides in total) and DNA vaccine VGX3100 (100μg/mice), all of which are currently in phase 2~3 clinical trials.

We hope that the above explanations and modifications will address your concerns.

Comments 6: Line 339 “67% (4/6) and 50% (3/6) of the mice remaining tumor-free up to 80 days post-tumor challenge.” You treated these animals three times for therapy. Would additional treatments increase the tumor-free survival in these mice?

Response 6: Thank you for your comment! Many studies have proposed that protein and peptide vaccines, although theoretically highly immunogenic, tend to be easily excreted by the body and have a short expiration date. It has been demonstrated in several previous studies that a primary-booster immunization regimen would help induce a more robust and long-lasting CTL response accompanied by the production of memory T cells [Vaccine. 2001;19(27):3652-3660.; Mol Immunol. 2015;64(2):295-305]. Therefore, in accordance with the protocol used in most preclinical trials for HPV16 vaccines, we adopted an immunization schedule of 3 doses immunized at one-week intervals between each dose. In our subsequent studies, we plan to further explore in the future whether fewer doses or dosages of vaccination can  can still confer adequate immunoprotection.

4. Response to Comments on the Quality of English Language

Point : Proofreading by a native English editor would help to increase the readership.

Response : Thank you for the suggestion. We have now worked on both language and readability and have also involved native English speakers for language corrections by MDPI (English-78884). We really hope that the flow and language level have been substantially improved.

5. Additional clarifications

Reviewer 5 Report

Comments and Suggestions for Authors

The study employed various immunoinformatics techniques to predict the HLA-I class-restricted CTL epitopes for the E5, E621, and E7 proteins of Human Papillomavirus (HPV) type 16. Through screening, conserved CTL epitopes E5, E6, and E7 were identified for human/mouse MHC restriction. Further, the selected epitope combinations were evaluated for immunogenicity in mice, and the antitumor effects of multi-epitope peptide vaccine E5E6E7pep11 and recombinant protein vaccine CTB-25 Epi11E567 were assessed in a mouse tumor model. The results showed that the mixed epitope peptide induced antigen-specific IFN-γ production in mice, providing 100% protection against tumor growth after prophylactic immunization. Additionally, both multi-epitope vaccines significantly inhibited tumor growth and prolonged survival in mice. In summary, this study successfully designed and evaluated a multi-epitope vaccine targeting the E5, E6, and E7 proteins of HPV16, demonstrating its potential immunogenicity and antitumor effects, offering a promising strategy for the immunotherapy of HPV-associated tumors.

Minor issues:

  1. The resolution of Figure 1 and Figure 2 is low.
  2. Does the construction of the vaccine with only CTL epitopes imply that helper T lymphocytes play no or only a minor role in HPV prevention and treatment?
  3. Why are allergenic epitopes included in Table 1?
  4. The selected antigenic epitopes were predicted for allergenicity but not for toxicity-related predictions. Are there any safety concerns?
  5. The mouse tumor model used in this study does not express the E5 oncoprotein on the surface; the antitumor efficacy should be further validated in a tumor model expressing the E5 protein.

Author Response

Response to Reviewer 5 Comments

1. Summary

Thank you very much for taking the time to review this manuscript. Please find the detailed responses below and the corresponding revisions changes in the re-submitted files.

2. Questions for General Evaluation

Reviewer’s Evaluation

Response and Revisions

Does the introduction provide sufficient background and include all relevant references?

Can be improved

Further additions have been made.

Are all the cited references relevant to the research?

Can be improved

Corrections and refinements have been made

Is the research design appropriate?

Yes

Are the methods adequately described?

Yes

Are the results clearly presented?

Can be improved

Further additions have been made.

Are the conclusions supported by the results?

Can be improved

3. Point-by-point response to Comments and Suggestions for Authors

Comments 1: The study employed various immunoinformatics techniques to predict the HLA-I class-restricted CTL epitopes for the E5, E621, and E7 proteins of Human Papillomavirus (HPV) type 16. Through screening, conserved CTL epitopes E5, E6, and E7 were identified for human/mouse MHC restriction. Further, the selected epitope combinations were evaluated for immunogenicity in mice, and the antitumor effects of multi-epitope peptide vaccine E5E6E7pep11 and recombinant protein vaccine CTB-25 Epi11E567 were assessed in a mouse tumor model. The results showed that the mixed epitope peptide induced antigen-specific IFN-γ production in mice, providing 100% protection against tumor growth after prophylactic immunization. Additionally, both multi-epitope vaccines significantly inhibited tumor growth and prolonged survival in mice. In summary, this study successfully designed and evaluated a multi-epitope vaccine targeting the E5, E6, and E7 proteins of HPV16, demonstrating its potential immunogenicity and antitumor effects, offering a promising strategy for the immunotherapy of HPV-associated tumors.

Response 1: We feel great thanks for your professional review work on our article.

Comments 2: The resolution of Figure 1 and Figure 2 is low.

Response 2: I apologize for the low resolution of Figures 1 and 2. We have provided higher-resolution versions of these figures in the revised manuscript. Thank you for bringing this to our attention.

Comments 3: Does the construction of the vaccine with only CTL epitopes imply that helper T lymphocytes play no or only a minor role in HPV prevention and treatment?

Response 3: Vaccines focusing on CTL epitopes aim to generate a strong cytotoxic response against the targeted virus or tumor cells. This strategy does not negate the importance of helper T cells but rather focuses on a specific aspect of the immune response that is critical for the direct elimination of infected or malignant cells. Our current results indicate that the targeted presentation of CTL epitopes effectively inducted the secretion of antigen-specific IFN-γ, which is essential for anti-tumor responses and combating HPV infections, as well as preventing and treating the onset and progression of tumors. However, as outlined in the last paragraph of the Discussion, our study does not delve into the immune regulatory mechanisms against tumors and long-term immune defense. Helper CD4 T cells play a crucial role in enhancing the proliferation of CTL clones and guiding their transformation into effector and memory cells. Therefore, incorporating Th epitopes may direct the vaccine-induced anti-tumor response towards sustainable and comprehensive regulation. Based on the current study, we are trying to further predict Th epitopes and improve vaccine design to maximize immunoprotection and therapeutic efficacy of multi-epitope vaccines. (Supplementary to the Discussion section of the manuscript - page 13, Phase 3, line 480–483, highlighted).

Comments 4: Why are allergenic epitopes included in Table 1?

Response 4: Thank you for pointing this out. For the reason outlined below, we incorporated three potential sensitizing epitopes into the vaccine design process. Firstly, the allergenicity of proteins or peptides is determined by their complex processing, presentation, and interaction with the immune system within the host body. Several factors can affect the allergenic potential of peptide mixtures, including the proportion of various components in the mixture, peptide processing and presentation, as well as the induction of immune tolerance. When peptide combinations are mixed and administered into the body, interactions between peptides may affect their overall allergenic potential. Additionally, mixing multiple peptides may dilute potential allergenic epitopes throughout the composition, thereby reducing allergenicity caused by individual peptides. Hence, multi-peptide vaccines may not necessarily be allergenic. During continuous observation of mice vaccinated with these preparations, no allergic-related characteristics were observed in the mice. Therefore, we opted to retain a small number of potentially allergenic peptides to ensure epitope diversity, and to ensure the vaccine can elicit a sufficient immune response to provide protection.

Comments 5: The selected antigenic epitopes were predicted for allergenicity but not for toxicity-related predictions. Are there any safety concerns?

Response 5: Thank you very much for the reminder. We have performed toxicity predictions, but did not show the results in previous manuscripts. Toxin Server predictions showed that all 11 candidate epitopes were non-toxic, and we have added the results to Table 1.

Comments 6: The mouse tumor model used in this study does not express the E5 oncoprotein on the surface; the antitumor efficacy should be further validated in a tumor model expressing the E5 protein.

Response 5: Thank you for pointing this out. We fully agree with your comments,and indeed we are making further efforts in this regard. Further validation of the protective effect on E5-expressing TC-1 tumour models would help to more comprehensively evaluate the efficacy of vaccines containing E5 antigenic epitopes, and we are actively working on this, but it will take some time. We have added a discussion of the limitations of the study (Supplemented in the Discussion section of the manuscript- - page 12, Phase 3, line 437–440, highlighted.)

4. Response to Comments on the Quality of English Language

Point : English language fine. No issues detected.

Response : Thank you for your careful review.

5. Additional clarifications

In response to the comments from all peer reviewers, we have incorporated additional content supplements and modifications into the manuscript, which are highlighted within the text.